# GC-MS-Olfactometric Characterization of Volatile and Key Odorants in Moringa (*Moringa oleifera*) and Kinkeliba (*Combretum micranthum* G. Don) Herbal Tea Infusions Prepared from Cold and Hot Brewing

Kouame Fulbert Oussou [1], Gamze Guclu [1], Onur Sevindik [2], Hasim Kelebek [2], Małgorzata Starowicz [3] and Serkan Selli [1,*]

[1] Department of Food Engineering, Faculty of Engineering, Cukurova University, Adana 01330, Turkey
[2] Department of Food Engineering, Faculty of Engineering, Adana Alparslan Turkes Science and Technology University, Adana 01250, Turkey
[3] Department of Chemistry and Biodynamics of Food, Institute of Animal Reproduction and Food Research Polish Academy of Sciences, Tuwima Street 10, 10-748 Olsztyn, Poland
[*] Correspondence: sselli@cu.edu.tr

**Abstract:** Herbal teas are a popular global drink and are widely used in many traditional medicines. Key odorants are one of the main parameters to elucidate the final herbal tea's overall quality and consumer acceptance. In the present study, for the first time, the brewing effect on volatile, key odorants, total phenolic contents, and antioxidant potential of Moringa (*Moringa oleifera*) and Kinkeliba (*Combretum micranthum* G. Don) herbal teas was comprehensively investigated. Two different infusions were studied and labeled as 25 °C/24 h (room temperature) and 98 °C/10 min (hot temperature). A total of 45 and 44 aroma compounds were detected in Moringa cold and hot teas, respectively, whereas 39 volatiles were determined in both infusion techniques for Kinkeliba herbal teas. The total amount of volatile compounds in both cold-infused herbal teas was higher than those in the hot-infused ones. Based on GC-MS-Olfactometry results, 19 and 21 key odorants in total were found in Moringa and Kinkeliba teas, respectively. The principal key odorants in Moringa teas with the highest flavor dilution (FD) factors were (*E*)-2-hexen-1-ol (herbal/fresh aroma), 3-hexanol (green/grassy), 2-phenyl ethanol (floral/rose), while in Kinkeliba teas they were 2-hexanol (herbal/green) and 3-penten-2-ol (green/fresh). The total phenolic content and antioxidant potential in Moringa and Kinkeliba teas increased using hot infusion. Principal component analysis showed that each tea infusion was clearly discriminated in terms of its volatile profiles. Our findings demonstrated that the brewing procedures had a significant impact on the key odorants of Moringa and Kinkeliba infusions.

**Keywords:** Moringa; Kinkeliba; herbal tea; key odorants; antioxidant; total phenolics; brewing

## 1. Introduction

Moringa (*Moringa oleifera* Lam.) is the most popular crop and is one of 13 known species from the Moringaceae family [1]. It is a fast-growing and attractive tree commonly named as "miracle tree" because of its tremendous nutritious and therapeutic potencies. It is widely cultivated and distributed in tropical and subtropical areas [2], especially in northeast India, Sri Lanka, and Africa. Moringa is a perennial plant 10–12 m in height, with creamy-white flowers and light-green foliage. In western Africa, the different morphological parts of Moringa, such as leaves, roots, bark, flowers, stems, and seeds, are considered precious nutritive and phytochemical sources, and they are extensively used as foodstuff, livestock feed, and for medicinal purposes. The cold and hot infused beverage of the dried leaves of Moringa is used as flavoring, coloring, and preservative

agents in traditional fermented drinks and other foodstuffs [3,4]. Recently, clinical investigations proved that Moringa had enormous and effective health properties including anti-inflammatory, hyperglycemic alternative, weight regulation, anti-tumor, cardiac and circulatory stimulant, anti-diabetic, antipyretic, anti-epileptic, anti-ulcer, anti-bacterial, anti-fungal, anti-hypertensive, cholesterol-lowering and antioxidant activities [3–5]. It is worth noting that these biological properties stem from the positive effects of phytochemical compounds such as phenolics contained in different parts of Moringa [6,7].

Kinkeliba *(Combretum micranthum* G. Don) commonly called "long life herbal tea" or "plant to heal" belongs to the Combretaceae family and can be found throughout tropical regions, where it is extensively used in traditional medicine [8]. Kinkeliba is an undomesticated bushy shrub or small tree approx. 5 m in size and found in west Africa, mostly in Ivory Coast, Mali, Guinea, Burkina Faso, Ghana, Togo, Nigeria, Gambia, Benin, and Senegal [9–11]. The leaves of Kinkeliba are widely consumed by the west African populations as a herbal tea due to their ethnomedical properties such as a tranquilization, tonic, digestive, and diuretic [9,11]. Ethnobotanical and ethnopharmacological studies on Kinkeliba revealed its beneficial effects on humans including blood sugar and cholesterol level reduction, anti-hypertensive, anti-inflammatory, neuro and nephron-protective, anti-malarial, anti-bacterial, anti-fungal, anti-viral and antioxidant [8,12–15]. Kinkeliba contains various phenolic compounds such as flavonoids and phenolic acids, mainly gallic acid, myricetin-3-*O*-rutinoside, rutin trihydrate, orientin, catechin, vitexin, quercitrin, and benzoic acid [10,13,16], which are responsible for the observed biological effects. In addition, Kinkeliba herbal tea is used for its unique and characteristic aroma and flavor properties [17].

The sensory properties of foods and beverages are the result of the production of a large number of non-volatile (e.g., phenolic compounds, amino acids, proteins, sugar, lipids, salt, etc.) and volatile compounds derived from various metabolites [18]. These compounds of low molecular weight (below 300 Da) are a complex mixture of volatile components readily vaporized at room temperature. The volatile compounds can be lost (by degradation or evaporation) or transformed into new volatile molecules owing to chemical reactions [19]. These transformations in food products may affect the quality of the final products and the consumers' acceptance. Moreover, the brewing techniques applied to the preparation of herbal infusions have been found to induce significant changes in the aroma composition of herbal infusions [20,21]. The volatile profile of most of the different tea and herbal tea samples contains a massive number of compounds, but only a small part of these compounds play a role in the formation of characteristic aromas, also known as key odorant compounds. Olfactometric analyzes are used to identify these aroma-active compounds [22], which have been successfully applied to many herbal teas including Iranian endemic borage *(Echium amoenum)* [20], Beninese roselle *(Hibiscus sabdariffa* L.) infusions [21], and Turkish thyme *(Thymus vulgaris* L.) tea [23] infusions.

Although Kinkeliba and Moringa teas are extensively consumed, to the best of our knowledge, no research has been conducted on their aroma and key aroma compounds. Therefore, the present study investigated the key odorants of Kinkeliba and Moringa teas produced from two different infusion techniques. The aroma compounds were extracted using the liquid-liquid extraction (LLE) technique and identified by gas chromatography coupled with mass spectrometry (GC-MS). The key odorants of the samples were determined using aroma extract dilution analysis (AEDA) and GC-MS-olfactometry (GC-MS-O) analyses. In addition, the total phenolic content and antioxidant potential of the infusions were determined.

## 2. Materials and Methods

### 2.1. Plant Materials and Reagents

The leaves of Moringa (approx. 1.5 kg) and Kinkeliba (approx. 1.5 kg) were collected from Abomey-Calavi/Benin Republic in 2020. The plant variety was identified and a voucher specimen (AA66/1645/HNB for Moringa and YH 356/HNB for Kinke-

liba) deposited leaves at the Experimental Botanic Garden of the University of Abomey-Calavi/Faculty of Agriculture. The botanical identification of the collected samples was carried out at this facility. The leaves were shade-dried for seven days, packed in brown bottles with screw caps, and stored in the fridge (4 ± 2 °C) for volatile analysis. The water used for analysis was purified using a Millipore-Q system (Millipore Billerica, MA, USA). Distilled dichloromethane (≥99.9% purity for GC) was used as a solvent for volatile compound extraction. Dichloromethane and other chemicals were purchased from Merck (Darmstadt, Germany) company. All standards of volatile compounds and other chemical compounds were purchased from Sigma Aldrich (St. Louis, MO, USA).

*2.2. Samples Infusion Preparation*

The tea infusion preparation process applied to both samples was as follows: (1) for hot infusion, 3 g of each dried leaves sample was blended with 200 mL of hot (98 ± 2 °C) distilled water for 10 min; (2) for cold infusion, 3 g of each dried leaves were drenched in 200 mL of distilled water for 24 h at 25 °C. A scientific water bath (Daihan WB-11, Malaysia) was used to control the temperature infused in both samples. After completion of each infusion, the hot tea samples were immediately cooled to room temperature.

*2.3. Total Soluble Solids, Color, and pH Determination*

The total soluble solids (°Brix) of the tea samples were measured using an Abbe refractometer (Westover, Woodinville, WA, USA). Analysis of the colors of the tea infusions was carried out using a Hunter Lab scan colorimeter (Reston, VA, USA). The color system profile ($L^*$ $a^*$ $b^*$) was measured using a Commission Internationale de l'Eclairage system [24]. Tea infusions were placed into the sample cell (6 cm diameter; 2 cm deep) and the data were recorded. The pH of the samples was directly determined using a pH meter (ORION 3-Star, Waltham, MA, USA).

*2.4. Analysis of Total Phenolic Content*

The total phenolic contents of the herbal tea samples were evaluated by the Folin–Ciocalteu method with some modifications [25,26]. Here, 1 mL of the phenolic extract, 60 mL of distilled water, 5 mL of Folin–Ciocalteu reagent, and 15 mL of $Na_2CO_3$ solution (20%) were mixed in a 100 mL flask. Then, the mixture was incubated for 2 h in the dark at room temperature and the absorbance was measured at 765 nm using a UV-Visible spectrophotometer (Schimadzu-1201, Kyoto, Japan). Blank solution samples were prepared under the same conditions as mentioned above, replacing the plant extract with purified water. The data were expressed as mg gallic acid equivalent per liter (mg GAE/L), and calibrated using a gallic acid standard curve (50–500 ppm).

*2.5. Analysis of Antioxidant Properties*

The antioxidant potential of tea samples was assessed using DPPH (1,1-diphenyl-2-picrylhydrazyl) and ABTS (2,2′-azinobis-(3-ethylbenzothiazoline-6-sulfonic acid) assays according to Rahim et al. [27]. The absorbance values of DPPH and ABTS were measured at 515 nm and 734 nm, respectively. The measurements were performed using a UV-Visible spectrophotometer (Schimadzu-1201, Japan) and the results were presented as mmol Trolox equivalent per liter (mmol TE/L).

*2.6. Liquid-Liquid Extraction of Volatile Compounds*

Liquid-liquid extraction technique was employed to extract the aroma compounds of the herbal tea samples. Briefly, the extraction procedure was conducted using 100 mL of the tea samples with the addition of 5 μL of 4-nonanol (41.5 μg/L) as internal standard and 150 mL of dichloromethane solvent with high purity in a magnetic mixer at 4–5 °C under a nitrogen atmosphere for 60 min, followed by centrifugation (9000× *g*) at 4 °C for 15 min. The aromatic extract was dried over $Na_2SO_4$ and concentrated to 0.5 mL on the

"Vigreux" distillation column at 45 °C. Extractions were performed in triplicate and three independent extracts for each herbal tea infusion were accomplished [28].

### 2.7. GC-FID and GC-MS Analysis of the Tea Samples

Gas chromatography with a flame ionization detector (GC-FID) (6890N, Agilent, Wilmington, DE, USA) was utilized for the quantification of aroma constituents. Separation of the compounds was accomplished via a DB-WAX capillary column (60 m × 0.25 mm × 0.4 μm). The column temperature was programmed as constant at 60 °C for 3 min, then increased to 220 °C with a rate of 2 °C/min and then elevated to 245 °C at a rate of 3 °C/min, and held at this value for 20 min. Then, 3 μL of the extract was injected into the GC injection port. Helium gas was utilized as the carrier with a flow rate of 1.5 mL/min. A temperature of 250 °C was set for the injector and detector. Mass spectrometry (5975B VL MSD, Agilent, DE, USA) based on GC was employed for the detection of the aroma constituents. Scanning was carried out at 29–350 mass/charge (m/e) at one-second intervals with 70 eV MS ionization energy, 250 °C ion source, and 120 °C quadrupole temperatures. The peaks were identified by injecting the standard solutions of the compounds. Mass spectrums of the non-standard constituents were assessed by evaluating the mass spectra of the aroma libraries available on the computer (Wiley 11.0, NIST-98, and Flavor.2L). After the assignment of the peaks, the quantities of the aroma substances were calculated by the internal standard procedure. Then, 4-Nonanol was used as the internal standard for aroma analysis since it fulfilled all significant criteria as internal standards and behaved similarly to the main groups of analytes in the studied tea samples. A quantitative method based on a combination of experimental calibration by internal standards and FID response factors was used for the quantification of each aroma compound. [21,23]. Each analysis was performed in triplicate.

### 2.8. Aroma Dilution Extract Analysis (AEDA)

The key odorants of the Kinkeliba and Moringa infusions were appraised by two qualified sniffers using a GC-MS-O instrument. The sniffing procedure of the extracts was accomplished in two sessions (23 min each). The AEDA technique was implemented to evaluate the flavor dilution (FD) parameters of the aroma-active components based on previous research [20,21]. Dichloromethane was used to sequentially dilute the samples in a ratio of 1:1, 1:2, 1:4, 1:8, 1:16, . . . , 1:2048. Sniffing was completed until no odor was detected.

### 2.9. Sensory Analysis

The sensory analysis involved 11 descriptors, including aroma, color, sweet, fruity, astringency, herbal, green, floral, overall taste, bitterness, and general acceptance, which were previously defined by the nine panelists (four females and five males aged 25 to 50 years old). Two infusion samples of each herbal tea were given to the panelists and for each descriptor, these were instructed to determine the intensity on a 100 mm scale [29].

### 2.10. Statistical Analysis

Experimental data were evaluated by the analysis of variance (ANOVA) in SPSS software (V.20.0; IBM Inc., Chicago, IL, USA). Also, the XLSTAT program (Addinsoft, New York City, NY, USA) was utilized for the principle component analysis (PCA) to examine the relationship between the volatile compounds and infusion techniques.

## 3. Results and Discussions

### 3.1. Colour, Total Phenolic, and Antioxidant Properties of the Samples

The chemical characteristics of Kinkeliba and Moringa infusions are listed in Table 1, where the brewing conditions significantly affected the color properties, pH, and total soluble solids of both tea samples ($p < 0.05$). The color property of tea, together with its aroma and taste is an important quality factor that mostly impacts consumers' choices and acceptance. As shown in Table 1, L * and b * values were higher in cold brewing than in hot brewing for both herbal teas. Cold-brewed teas possessed a lighter, yellow color. The dark

color of hot brewed samples could be due to the amount of polyphenol compounds released during the brewing process. Additionally, hot-brewed Moringa (1664.8 mg GAE/L) and Kinkeliba (1913.9 mg GAE/L) infusions had higher levels of total phenolics compared to the cold-brewed samples. These results suggested that thermal treatment during brewing may increase the release of phenolics. These findings were consistent with the other reported studies [21,30]. The antioxidant potential of the infusion from both samples was evaluated through DPPH and ABTS methods. Significant differences ($p < 0.05$) were observed between the antioxidant potentials of infusions obtained by cold (25 °C/24 h) and hot brewing (98 °C/15 min) methods. Furthermore, it was found that hot-brewed teas had higher antioxidant capacity due to their phenolic content. ABTS values of the samples were significantly higher than DPPH values in all tea infusions (Table 1). The observed differences may be due to ABTS solubility in a hydrophilic or lipophilic medium, while DPPH is more attractive in a lipophilic and partial lipophilic medium.

**Table 1.** Total phenolics compounds, antioxidant capacity, and general characteristics of tea infusion samples.

| | Moringa Tea | | Kinkeliba Tea | |
|---|---|---|---|---|
| **Contents *** | **25 °C/24 h** | **98 °C/10 min** | **25 °C/24 h** | **98 °C/10 min** |
| Total phenolic (mg GAE/L) | 1040.0 ± 16 [b] | 1664.8 ± 24 [a] | 1625.8 ± 28 [d] | 1913. 9 ± 45 [c] |
| ABTS (mmol TE/L) | 456.1 ± 27 [b] | 714.1 ± 31 [a] | 168.4 ± 31 [d] | 376.6 ± 37.4 [c] |
| DPPH (mmol TE/L) | 306.3 ± 33 [b] | 386.14 ± 27 [a] | 134.1 ± 2 [d] | 352.2 ± 23.1 [c] |
| Total soluble solid (°Brix) | 1.6 ± 0.1 [b] | 2.9 ± 0.1 [a] | 1.7 ± 0.1 [d] | 3.2 ± 0.1 [c] |
| pH | 6.2 ± 0.1 [a] | 5.6 ± 0.1 [a] | 6.2 ± 0.1 [c] | 6.2 ± 0.1 [c] |
| L * | 79.6 ± 0.1 [a] | 45.2 ± 0.1 [b] | 86.2 ± 0.1 [c] | 45.0 ± 0.1 [d] |
| a * | −6.7 ± 0.1 [b] | −1.7 ± 0.1 [a] | −4.7 ± 0.1 [d] | −2.9 ± 0.1 [c] |
| b * | 24.7 ± 0.1 [a] | 20.6 ± 0.1 [b] | 43.53 ± 0.02 [c] | 19.0 ± 0.1 [d] |

* Mean of three repetitions (n = 3) lowercase letters (a,b,c,d) in the same line of two different treatments of the same samples show a significant difference ($p < 0.05$) between the tea infusion technique applied.

### 3.2. Aroma Profiles of Tea Samples

The aroma profiles of Moringa and Kinkeliba tea infusions are reported in Table 2, where the linear retention index (LRI) value on the DB-Wax column and its concentrations (μg/L) are presented as means and standard deviation. A total of 44 aroma compounds were detected in Moringa tea infused at a hot temperature, which were classified as acids (9), furans (4), alcohols (10), aldehydes (6), ketones (3), lactone (1), pyrroles (2), terpenes (2), volatile phenols (5), and ester (1). A total of 43 compounds were identified in Moringa tea brewed at a cold temperature, which were classified as acids (9), furans (4), alcohols (9), aldehydes (6), ketones (3), lactone (1), pyrroles (2), terpenes (3), volatile phenols (5) and ester (1). It was found that the volatile compositions of both teas infused at hot and cold temperatures were similar. However, furfuryl alcohol and linalool were only found in the hot and cold brewed Moringa teas, respectively.

**Table 2.** Aroma profiles of Moringa and Kinkeliba herbal tea infusions.

| | | | Concentration (mean ± SD) [2] | | | | |
|---|---|---|---|---|---|---|---|
| | | | **Moringa Herbal Tea** | | **Kinkeliba Herbal Tea** | | |
| **No** | **Aroma Compounds** | **LRI [1]** | **98 °C/10 min** | **25 °C/24 h** | **98 °C/10 min** | **25 °C/24 h** | **Identification** |
| | Acids | | | | | | |
| 1 | Acetic acid | 1446 | 704.6 ± 16 [b] | 956.2 ± 16 [a] | 468.2 ± 14 [d] | 656.0 ± 12 [c] | LRI, MS, Std |
| 2 | Isovaleric acid | 1680 | 146.2 ± 58 [b] | 232.4 ± 44 [a] | nd | nd | LRI, MS, Std |
| 3 | Hexanoic acid | 1851 | 1288.5 ± 18 [b] | 2469.7 ± 29 [a] | 389.9 ± 10 [d] | 422.9 ± 13 [c] | LRI, MS, Std |
| 4 | Heptanoic acid | 1968 | 178.5 ± 2 [b] | 368.6 ± 76 [a] | 326.5 ± 7 [d] | 518.6 ± 3 [c] | LRI, MS, Std |

**Table 2.** *Cont.*

| No | Aroma Compounds | LRI [1] | Moringa Herbal Tea 98 °C/10 min | Moringa Herbal Tea 25 °C/24 h | Kinkeliba Herbal Tea 98 °C/10 min | Kinkeliba Herbal Tea 25 °C/24 h | Identification |
|---|---|---|---|---|---|---|---|
| | | | Concentration (mean ± SD) [2] | | | | |
| 5 | Octanoic acid | 2060 | 489.6 ± 10 [b] | 862.0 ± 27 [a] | 270.7 ± 27 [d] | 473.1 ± 46 [c] | LRI, MS, Std |
| 6 | Nonanoic acid | 2158 | 963.3 ± 77 [b] | 1476.4 ± 17 | 1824.9 ± 11 [d] | 3205.4 ± 89 [c] | LRI, MS, Std |
| 7 | Decanoic acid | 2314 | 193.3 ± 13 [b] | 369.8 ± 16 [a] | 106.8 ± 7 [d] | 348.6 ± 12 [c] | LRI, MS, Std |
| 8 | Phenylacetic acid | 2553 | 88.8 ± 11 [b] | 174.4 ± 18 [a] | nd | nd | LRI, MS, Std |
| 9 | Tetradecanoic acid | 2716 | 274.3 ± 14 [b] | 392.1 ± 15 [a] | 207.9 ± 9 [d] | 440.5 ± 12 [c] | LRI, MS, Std |
| | Total | | 4327.1 | 7301.6 | 3594.9 | 6065.1 | |
| | Furans | | | | | | |
| 10 | 2,5-Dimethylfuran | 958 | 176.3 ± 16 [a] | 357.5 ± 24 [b] | 96.1 ± 10 [c] | 56.0 ± 7 [d] | LRI, MS, Std |
| 11 | Furfural | 1469 | 2281.0 ± 54 [a] | 1046.5 ± 86 [b] | 2918.3 ± 68 [c] | 1635.3 ± 15 [d] | LRI, MS, Std |
| 12 | 5-Methyl 2(5H)-furanone | 1664 | 1789.7 ± 64 [a] | 962.3 ± 24 [b] | 2613.6 ± 10 [c] | 1073.7 ± 15 [d] | LRI, MS, Std |
| 13 | 5-Hydroxy-2(3H) benzofuranone | 2325 | 843.2 ± 12 [a] | 551.5 ± 22 [b] | 919.7 ± 29 [c] | 475.1 ± 20 [d] | LRI, MS, Std |
| 14 | 2,3-Dihydro-benzofuran | 2337 | nd | nd | 522.1 ± 25 [c] | 350.4 ± 11 [d] | LRI, MS, Std |
| | Total | | 5090.2 | 2917.8 | 7099.8 | 3590.5 | |
| | Alcohols | | | | | | |
| 15 | 1-Penten-3-ol | 1158 | nd | nd | 3234.8 ± 82 [d] | 6688.5 ± 71 [c] | LRI, MS, Std |
| 16 | 4-Methyl-2-pentanol | 1181 | 145.0 ± 11 [b] | 474.2 ± 24 [a] | nd | nd | LRI, MS, Std |
| 17 | 3-Penten-2-ol | 1182 | 765.3 ± 31 [b] | 936.5 ± 66 [a] | 3268.2 ± 88 [d] | 5962.5 ± 215 [c] | LRI, MS, Std |
| 18 | 3-Hexanol | 1190 | 1541.6 ± 11 [b] | 2345.4 ± 12 [a] | 2236.5 ± 96 [d] | 4754.1 ± 130 [c] | LRI, MS, Std |
| 19 | 2-Hexanol | 1313 | 396.4 ± 19 [b] | 626.6 ± 15 [a] | 23,641.4 ± 186 [d] | 32,728.1 ± 274 [c] | LRI, MS, Std |
| 20 | (E)-3-Hexen-1-ol | 1384 | 2846.4 ± 81 [b] | 4463 ± 124 [a] | 1817.4 ± 7 [d] | 2032.0 ± 12 [c] | LRI, MS, Std |
| 21 | (Z)-3-Hexen-1-ol | 1401 | 2667.3 ± 133 [b] | 4135.6 ± 109 [a] | 2043.6 ± 83 [d] | 3410.8 ± 56 [c] | LRI, MS, Std |
| 22 | 2,3-Butanediol | 1548 | 428.5 ± 6 [b] | 285.6 ± 25 [a] | 927.4 ± 45 [d] | 2044.6 ± 108 [c] | LRI, MS, Std |
| 23 | Furfuryl alcohol | 1661 | 88.9 ± 1 [b] | nd | 263.4 ± 2 [a] | nd | LRI, MS, Std |
| 24 | 2-Phenyl ethanol | 1867 | 6189.6 ± 91 [b] | 9692.4 ± 57 [a] | 5261.2 ± 62 [d] | 6727.4 ± 21 [c] | LRI, MS, Std |
| 25 | Benzyl alcohol | 1877 | 841.3 ± 77 [b] | 1125.7 ± 101 [a] | 3601.9 ± 75 [d] | 4456.4 ± 10 [c] | LRI, MS, Std |
| | Total | | 15,910.3 | 24,085.0 | 46,295.8 | 68,804.4 | |
| | Aldehydes | | | | | | |
| 26 | (E)-2-Pentenal | 1147 | 92.8 ± 1 [b] | 233.5 ± 4 [a] | 386.4 ± 31 [d] | 706.3 ± 26 [c] | LRI, MS, Std |
| 27 | 3-Methyl-2-butenal | 1215 | 187.1 ± 2 [b] | 468.2 ± 6 [a] | nd | nd | LRI, MS, Std |
| 28 | (E)-2-Hexenal | 1219 | 906.6 ± 26 [b] | 2689.1 ± 16 [a] | 2447.2 ± 44 [d] | 4233.1 ± 66 [c] | LRI, MS, Std |
| 29 | (E)-2-Heptenal | 1334 | 61.8 ± 2 [b] | 88.0 ± 8 [a] | nd | nd | LRI, MS, Std |
| 30 | Benzaldehyde | 1508 | 174.5 ± 7 [b] | 303.3 ± 5 [a] | 106.2 ± 2 [d] | 386.5 ± 5 [c] | LRI, MS, Std |
| 31 | Phenylacetaldehyde | 1648 | 120.6 ± 5 [b] | 274.2 ± 4 [a] | nd | nd | LRI, MS, tent |
| | Total | | 1543.4 | 4056.3 | 2939.8 | 5325.9 | |
| | Ketones | | | | | | |
| 32 | 3-Methyl 2 pentanone | 1016 | nd | nd | 767.5 ± 16 [d] | 1076.6 ± 15 [c] | LRI, MS, Std |
| 33 | Propiophenone | 1737 | 816.2 ± 11 [b] | 1022.9 ± 44 [a] | 1192.6 ± 55 [d] | 3921.5 ± 89 [c] | LRI, MS, tent |
| 34 | Piperitenone | 1905 | nd | nd | 4936.6 ± 86 [d] | 6877.5 ± 109 [c] | LRI, MS, tent |
| 35 | 3-Hydroxy-2-butanone | 2287 | 2163.9 ± 29 [b] | 3859.5 ± 64 [a] | 989.5 ± 13 [d] | 2197.1 ± 66 [c] | LRI, MS, Std |
| 36 | 3-Hydroxy-β-ionone | 2700 | 66.3 ± 6 [b] | 114.1 ± 6 [a] | nd | nd | LRI, MS, Std |

**Table 2.** *Cont.*

| No | Aroma Compounds | LRI [1] | Moringa Herbal Tea 98 °C/10 min | Moringa Herbal Tea 25 °C/24 h | Kinkeliba Herbal Tea 98 °C/10 min | Kinkeliba Herbal Tea 25 °C/24 h | Identification |
|---|---|---|---|---|---|---|---|
| | | | \multicolumn Concentration (mean ± SD) [2] | | | | |
| | Total | | 3046.4 | 4996.5 | 7886.2 | 14,072.7 | |
| | Lactone | | | | | | |
| 37 | γ-Butyrolactone | 1628 | 108.0 ± 2 [b] | 436.0 ± 12 [a] | 566.7 ± 5 [d] | 717.3 ± 8 [c] | LRI, MS, Std |
| | Total | | 108.0 | 436.0 | 566.7 | 717.3 | |
| | Pyrroles | | | | | | |
| 38 | 2-Acetylpyrrole | 1949 | 1464.0 ± 14 [a] | 848.4 ± 35 [b] | 397.9 ± 5 [c] | 188.1 ± 10 [d] | LRI, MS, tent |
| 39 | 2-Pyrrolidinone | 2037 | 501.8 ± 12 [a] | 347.1 ± 8 [b] | 2466.2 ± 16 [c] | 986.5 ± 3 [d] | LRI, MS, Std |
| | Total | | 1965.8 | 1195.5 | 2864.1 | 1174.6 | |
| | Terpenes | | | | | | |
| 40 | β-Myrcene | 1165 | nd | nd | 88.7 ± 1 [d] | 106.4 ± 5 [c] | LRI, MS, Std |
| 41 | *dl*-Limonene | 1206 | 173.5 ± 7 [b] | 427.0 ± 11 [a] | nd | nd | LRI, MS, Std |
| 42 | β-Citronellal | 1489 | nd | nd | 142.5 ± 6 [d] | 287.5 ± 4 [c] | LRI, MS, Std |
| 43 | Carvacrol | 1547 | nd | nd | nd | 66.6 ± 1 | LRI, MS, Std |
| 44 | Linalool | 1565 | nd | 85.5 ± 1 | nd | nd | LRI, MS, Std |
| 45 | Thymol | 2167 | 52.3 ± 4 [b] | 71.0 ± 5 [a] | nd | nd | LRI, MS, Std |
| | Total | | 225.8 | 583.5 | 231.2 | 460.5 | |
| | Phenols | | | | | | |
| 46 | Phenol | 1958 | 931.5 ± 14 [b] | 1049.5 ± 13 [a] | 1984.6 ± 48 [c] | 3728.0 ± 62 [d] | LRI, MS, Std |
| 47 | *p*-Cresol | 2074 | 140.5 ± 4 [a] | 328.7 ± 2 [b] | nd | nd | LRI, MS, tent |
| 48 | *p*-Vinylguaiacol | 2203 | nd | nd | 167.5 ± 6 [c] | 347.8 ± 10 [d] | LRI, MS, tent |
| 49 | 2,4-Di-tert-butylphenol | 2330 | 88.3 ± 6 [b] | 256.6 ± 24 [a] | nd | nd | LRI, MS, tent |
| 50 | Isoeugenol | 2266 | 169.5 ± 4 [b] | 445.3 ± 11 [a] | 848.4 ± 12 [c] | 1176.6 ± 9 [d] | LRI, MS, Std |
| 51 | Vanillin | 2540 | 224.0 ± 6 [b] | 576.4 ± 11 [a] | nd | nd | LRI, MS, Std |
| | Total | | 1553.8 | 2656.5 | 3000.5 | 5252.4 | |
| | Ester | | | | | | |
| 52 | Methyl 6-octadecenoate | 2531 | 181.1 ± 4 [b] | 296.9 ± 2 [b] | 322.6 ± 4 [d] | 571.6 ± 3 [c] | LRI, MS, tent |
| | Total | | 181.1 | 296.9 | 322.6 | 571.6 | |
| | General total | | 33,951.9 | 48,525.6 | 74,801.6 | 106,035.0 | |

[1] LRI: Linear retention index computed on DB-WAX capillary column. [2] Concentration: Mean of three replication as μg/L with different letters (a,b,c,d) in the same row of two different treatments of the same sample show significance ($p < 0.05$) between tea preparation methods applied, nd: not detected. Identification: methods of identification included LRI (linear retention index), MS tent (tentatively identified by MS), Std (identified by chemical standard).

Differences were observed in concentrations of the aforementioned volatiles (Table 2). To the best of our knowledge, the aroma compounds were identified and quantified for the first time in Moringa (*Moringa oleifera*) tea samples. In the case of Kinkeliba tea, a total of 38 volatile compounds were detected in hot-infused tea, which were classified as acids (7), furans (5), aldehydes (3), alcohols (10), ketones (4), pyrroles (2), lactone (1), terpene (2), volatile phenols (3), and ester (1), whereas in Kinkeliba cold brewed tea a total of 38 volatile compounds were determined, which were classified as acids (7), furans (5), aldehydes (3), alcohols (9), ketones (4), pyrroles (2), lactone (1), terpenes (3), volatile phenols (3), and ester (1). Total aroma concentration was observed to be significantly lower ($p < 0.05$) in Kinkeliba tea brewed at 98 °C/10 min (74,801.6 μg/L) than at 25 °C/24 h (106,035.0 μg/L). Previous reports have shown that most of these aroma compounds have been observed in different types of herbal teas [20,21,31,32].

*Alcohols*. The total amount of alcohols was 15,910.3 µg/L and 24,085.0 µg/L in hot and cold Moringa tea infusions, and 46,295.8 µg/L and 68,804.4 µg/L in hot and cold Kinkeliba tea infusions, respectively (Table 2). Alcohols were found to be quantitatively prominent aroma components in both tea samples. Among all the volatile alcohols present, 2-phenyl ethanol, *(E)*-3-hexen-1-ol, *(Z)*-3-hexen-1-ol, and 3-hexanol were determined as the main alcohol compounds in Moringa tea infusions, whereas in Kinkeliba tea infusions 2-hexanol, 2-phenyl ethanol, benzyl alcohol, 1-penten-3-ol, and 3-penten-2-ol were predominant. Reports have considered that alcohol constituents are generally formed from unsaturated fatty acids (linolenic and linoleic acids) through a series of enzymatic reactions such as lipoxygenase, hydroperoxide lyase, isomerase, and alcohol dehydrogenase actions [33]. In both tea samples, heat treatment resulted in a considerable decrease in the total amount of alcohol compounds (Table 2). Similar results have also been reported for Beninese roselle tea [21], where the amount of total alcohol compounds in cold roselle tea was 11.7 mg/L, while in the hot brew it was 5.43 mg/L.

*Acids.* A total of nine acid compounds were detected in Moringa hot and cold brewed teas, and seven acid compounds were detected in Kinkeliba samples. Isovaleric acid and phenylacetic acid were detected only in Moringa hot and cold brewed teas (Table 2). A significant number of these compounds have also been identified in Borage (*Echium amoenum*) herbal [20] and *Hibiscus sabdariffa* teas [30]. The obtained results showed that hexanoic acid was the dominant acid compound in Moringa samples, whereas nonanoic acid was in Kinkeliba teas. It is known that such compounds generally exhibit high odor threshold values, which slightly impact the overall aroma characteristics.

*Carbonyl compounds.* In the present study, six aldehydes (*(E)*-2-pentenal, 3-methyl-2-butenal, *(E)*-2-hexenal, *(E)*-2-heptenal, benzaldehyde, and phenylacetaldehyde) detected in Moringa tea samples; and three aldehydes (*(E)*-2-pentenal, *(E)*-2-hexenal, and benzaldehyde) were found in Kinkeliba tea samples. Their amounts were high in the cold brewing methods of both herbal tea samples (Table 2). Aldehydes are commonly found in herbal tea infusions and are part of an important aroma group that strongly influences the characteristic aroma profiles [21]. Among these aldehydes, *(E)*-2-hexenal was the most abundant in both samples and brewing methods in the present study. This unsaturated C6 aldehyde was recognized as the main aldehyde, which was an indicator of the degree of black tea fermentation of unsaturated fatty acids [34].

Three ketones, propiophenone, 3-hydroxy-2-butanone, and 3-hydroxy-$\beta$-ionone, were identified in both Moringa tea samples, while four ketones, 3-methyl-2 pentanone, piperitenone, propiophenone, and 3-hydroxy-2-butanone, were found in both Kinkeliba tea samples. As with other aroma groups, the amount of ketones was higher in cold-brewed teas. Schuh and Schieberle [35] showed that ketone compounds were a very important aroma chemical group in various herbal teas, providing characteristic floral and woody odor notes due to their low threshold values.

*Furans.* Furans are generally produced from the thermal processing of food, particularly from the Maillard reaction and ascorbic acid oxidation. In the present study, 2,5-dimethylfuran, furfural, 5-methyl 2(5H)-furanone, and 5-hydroxy-2(3H) benzofuranone were identified in Moringa cold and hot tea infusions. In addition to these compounds, 2,3-dihydrobenzofuran was detected in Kinkeliba teas (Table 2). Their amount increased significantly with the hot brewing method in both samples. Among the furans detected, furfural and 5-methyl 2(5H)-furanone were dominant in all tea samples (Table 2). Furthermore, such furan compounds were previously reported for different herbal tea samples [21,30].

*Terpenes.* The total amount of terpenes was quantified as 225.8 µg/L in hot brewed and 583.5 µg/L in cold brewed Moringa tea samples, whereas hot and cold brewed Kinkeliba herbal teas had terpenes in the amounts of 231.2 µg/L and 460.5 µg/L. Variations in terpene content may result from the biochemical composition of the raw materials and tea infusion conditions. *dl*-Limonene was the quantitatively dominant terpene in Moringa infusions, while $\beta$-citronellal was in Kinkeliba (Table 2). Terpene compounds including monoterpenes,

sesquiterpenes, and their oxygenated derivatives provide floral and green odors that are produced by terpenoid synthase enzyme activities [36].

*Volatile phenols.* In the case of volatile phenol constituents, phenol and isoeugenol were common in both herbal tea samples, while *p*-cresol, 2,4-di-tert-butylphenol and vanillin were only in Moringa tea infusions. However, *p*-vinyl guaiacol was detected only in Kinkeliba samples. Additionally, volatile phenol compounds were present in significant amounts in both herbal tea samples and their amount was approx. twice higher in Moringa tea samples (Table 2). In both herbal tea types, phenol was the most dominant volatile phenol and hot brewing significantly reduced its level ($p < 0.05$).

*Other volatile compounds.* Four other compounds, including one lactone ($\gamma$-butyrolactone), one ester (methyl 6-octadecenoate), and two pyrroles (2-acetylpyrrole and 2-pyrrolidinone), were detected in both types of herbal tea samples (Table 2).

Principal component analysis (PCA) was utilized to better visualize the effects of the two brewing methods on Moringa and Kinkeliba volatiles (Figures 1 and 2). A total of 44 variables (volatiles) were used for Moringa tea, and the observed variance was 100% (F1:75.04%, F2:24.96%). Moringa tea samples based on the infusion processes were differentiated into two separate groups in the PCA biplot (Figure 1). Moringa tea sample brewed at 98 °C/10 min was mostly characterized by phenol, furfural, 5-methyl-2(5H)-furanone, $\gamma$-butyrolactone, benzyl alcohol, and propiophenone variables. Moreover, the herbal tea infused at 25 °C/24 h was primarily characterized by 2-phenyl ethanol, 3-hexanol, nonanoic acid, *(Z)*-3-hexen-1-ol, hexanoic acid, *(E)*-3-hexen-1-ol and *(E)*-2-hexenal variables.

**Figure 1.** PCA biplot of aroma compounds in Moringa herbal tea.

Regarding Kinkeliba tea, 38 variables (volatiles) were utilized for PCA, and the observed variance was 100% (F1:65.09%, F2:34.91%) (Figure 2). The herbal tea brewed at 98 °C/10 min was characterized by furfural, 2-acetylpyrrole, 2-hexanol, 5-methyl 2(5H)-furanone, 2-phenyl ethanol, benzyl alcohol, and piperitenone variables. The herbal tea prepared at 25 °C/24 h was mainly characterized by 3-hexanol, 3-penten-2-ol, *(E)*-2-hexenal, propiophenone, phenol, nonanoic acid, and *(E)*-3-hexen-1-ol variables.

**Figure 2.** PCA biplot of aroma compounds in Kinkeliba herbal tea.

*3.3. Key Odorant Compounds in Moringa and Kinkeliba Herbal Teas Infusion*

　　A GC-MS-Olfactometry system associated with AEDA was used to elucidate the key odorants in aromatic extracts of Moringa and Kinkeliba herbal tea infusions. A total of 19 key odorants including alcohols (4), furans (2), acids (3), terpenes (2), aldehyde (1), volatile phenol (1), pyrrole (1), ketone (1), lactone (1), and three unknown aroma-active compounds were identified in Moringa tea cold and hot infusions, in which the flavor dilution (FD) factors ranged from 8 to 2048 and 4 to 2048, respectively. It was observed that all key odorants sensed in the olfactometry of cold and hot infusions of Moringa tea were similar in total number but varied significantly in their FD factors. These variations may be associated with the temperature and time differences in the infusion preparations, which may substantially impact the volatility and dominance of key aroma substances. The obtained results were in accordance with previous findings reported in the literature considering the effect of temperature and time factors on the aroma active profile of herbal tea infusions [20,21].

　　Alcohols were the main chemical group of Moringa tea infusions possessing the highest FD factors, which indicated that they impacted the overall key odorant profile. Among aroma-active alcohols, *(E)*-3-hexen-1-ol, 3-hexanol, and 2-phenyl ethanol were identified as the dominant odorants providing pleasant strong green/floral, green/floral, and floral/rose notes, respectively (Table 3). These aroma-active alcohols were previously determined in oolong, Longjing, green, Borage, and other herbal teas [20,37,38]. *(E)*-3-Hexen-1-ol, the most powerful aroma-active compound with the highest FD factors in both Moringa tea infusions (1024 and 2048 for hot and cold infusions, respectively), was also detected in Japanese green teas such as Sencha, Matcha, Gyokuro, and Hojicha [39].

　　Although *(E)*-2-hexenal was found as the only aroma-active aldehyde, its impact on the overall Moringa tea odor was quite substantial (FD = 512, FD = 1024 for hot and cold Moringa infusions, respectively) providing a green-grassy odor. According to Mao et al., this aldehyde was a key odorant of Congou black tea [40] and is known to be derived

from the oxidation of unsaturated fatty acids, which basically produce fatty acid-derived aroma compounds.

Two terpenes (*dl*-limonene and thymol) with citrusy/lemon and herbal/phenolic notes, volatile acids (acetic, hexanoic, and nonanoic acids) with vinegar, floral and green notes, and a lactone (γ-butyrolactone), a volatile phenol (isoeugenol), a ketone (3-hydroxy-2-butanone) were other key odorants of Moringa herbal tea. FD factors of all these aroma-active compounds were affected by the high-temperature preparation of Moringa herbal tea infusion.

Kinkeliba tea's key odorant profile was composed of a total of 19 key aroma compounds, including alcohols (6), furans (2), ketones (2), volatile acid, terpene, aldehyde, pyrrole, lactone, volatile phenol, and three unknown compounds (Table 4). All key odorants were found in both infusion types, but their FD factors ranged from 4 to 2048 and 4 to 512 in cold and hot infusions, respectively.

Similarly, Moringa tea's key odorant profile mostly consisted of alcohols, including 2-hexanol, 3-penten-2-ol, 1-penten-3-ol, *(E)*-3-hexen-1-ol and 2-phenyl ethanol, which were aroma active compounds identified in Kinkeliba tea infusions. Differently from Moringa tea, 2-hexanol was found to be the most powerful key odorant with the highest FD factors of 2048 and 512 in cold and hot infusion, respectively. This compound is known to exhibit a pleasant green scent and it has been previously addressed by Selli et al. [41] as the principal volatile compound of Roselle calyxes and by Amanpour et al. [20] in the Iranian Borage tea.

Three different aroma-active furans were detected, including furfural (caramel/pungent odor), 5-methyl-2(5H) furanone (burnt/roasted odor), and 5-hydroxy-2(3H) benzofuranone (burnt odor), in Kinkeliba tea infusions. Their flavor dilution factor ranged from 16 to 128 and the observed increase in FD value confirmed the heat effect on the key odorant profile of Kinkeliba tea infusion.

Another important class of key odorants of Kinkeliba teas was ketones, in which piperitenone (sweet, floral) possessed an FD value of 512 in cold and 128 in hot infusions. 3-Hydroxy-2-butanone was another ketone compound found in both types of Kinkeliba tea infusions. *(E)*-2-Hexenal was the only aldehyde but displayed lower FD factors (FD = 64 and FD = 256 for hot and cold infusions, respectively). Other classes of key odorants, such as volatile acids, terpenes, pyrroles, and unknown compounds, had lower FD values.

According to the data listed in Tables 3 and 4, the effect of high-temperature preparation of Moringa and Kinkeliba tea infusions significantly changed the dominancy of some marker notes, particularly the Maillard reaction-based compounds such as furans and pyrroles. In both tea samples, hot infusions possessed remarkably higher FD factors for furfural, 5-ethyl-2(5H)-furanone, 2-acetylpyrrole, 5-methyl-2(5H) furanone, and 5-hydroxy-2(3H) benzofuranone than the cold infusions. These key aroma compounds mostly provided caramel/pungent, nutty, burnt/roasted, burnt notes to the tea infusions. Apart from furans and pyrroles, another marker found at high temperature was unknown aroma-active compounds, which contributed popcorn-like and cooked vegetable notes to the overall scent of both herbal teas. This well-known popcorn-like odor was previously mentioned by Kumasawa and Masuda [42], who identified key aroma compounds in several green teas. Similarly, in the present study, a popcorn-like odor was provided by some unknown aroma-active compounds, which possessed low threshold values but the GC-MS instrument was not able to detect them due to their low concentrations.

Most of the aroma-active compounds detected in the present study were previously characterized as a potential key odorant for the overall aroma in several herbal teas such as Oolong, black, green, Borage and Roselle teas [20,21,37,43]. However, to the best of our knowledge, no publication provides any detailed information about aroma profiles and key odorants of Moringa and Kinkeliba herbal teas.

**Table 3.** Key odorants of Moringa herbal tea obtained from two brewing processes.

| No | [a] LRI | Compounds | Classes | Odor Description | [b] FD Factor ≥4 | |
|----|---------|-----------|---------|------------------|-------------------|---|
| | | | | | 98 °C/10 min | 25 °C/24 h |
| 1 | 1190 | 3-Hexanol | alcohol | green/floral | 512 | 1024 |
| 2 | 1206 | *dl*-Limonene | terpene | citrusy/lemon | 32 | 64 |
| 3 | 1219 | (*E*)-2-Hexenal | aldehyde | green/grassy | 512 | 1024 |
| 4 | 1384 | (*E*)-3-Hexen-1-ol | alcohol | green/floral | 1024 | 2048 |
| 5 | 1446 | Acetic acid | acid | vinegar | 16 | 32 |
| 6 | 1469 | Furfural | furan | caramel/pungent | 128 | 32 |
| 7 | 1628 | γ-Butyrolactone | lactone | caramel/creamy | 64 | 64 |
| 8 | 1664 | 5-Ethyl-2(5H) furanone | furan | burnt/caramel | 512 | 128 |
| 9 | 1851 | Hexanoic acid | acid | floral | 32 | 64 |
| 10 | 1867 | Unknown I | | green tea/herbal | 32 | 8 |
| 11 | 1867 | 2-Phenyl ethanol | alcohol | floral/rose | 512 | 1024 |
| 12 | 1877 | Benzyl alcohol | alcohol | leaf/floral | 32 | 64 |
| 13 | 1949 | 2-Acetylpyrrole | pyrrole | nutty | 256 | 64 |
| 14 | 2017 | Unknown II | | popcorn-like | 64 | 8 |
| 15 | 2158 | Nonanoic acid | acid | green | 8 | 32 |
| 16 | 2167 | Thymol | terpene | herbal/phenolic | 4 | 8 |
| 17 | 2266 | Isoeugenol | phenol | phenolic | 8 | 64 |
| 18 | 2287 | 3-Hydroxy-2-butanone | ketone | cheesy/fatty | 8 | 16 |
| 19 | 2442 | Unknown III | | floral/ fresh | 64 | 16 |

[a] Linear retention index (LRI) calculated on DB-WAX capillary column. [b] FD factor of the odorant compound was determined by AEDA.

**Table 4.** Aroma-active compounds of Kinkeliba herbal tea obtained from two infusion processes.

| No | [a] LRI | Compounds | Classes | Odor Description | [b] FD Factor ≥4 | |
|----|---------|-----------|---------|------------------|-------------------|---|
| | | | | | 98°/10 min | 25°/24 h |
| 1 | 1158 | 1-Penten-3-ol | alcohol | green/grassy | 256 | 512 |
| 2 | 1182 | 3-Penten-2-ol | alcohol | herbal/green | 512 | 1024 |
| 3 | 1196 | Unknown I | | popcorn like | 16 | 4 |
| 4 | 1219 | (*E*)-2-Hexenal | aldehyde | green/grassy | 64 | 256 |
| 5 | 1313 | 2-Hexanol | alcohol | green | 512 | 2048 |
| 6 | 1384 | (*E*)-3-Hexen-1-ol | alcohol | green/floral | 128 | 256 |
| 7 | 1469 | Furfural | furan | caramel/pungent | 128 | 64 |
| 8 | 1489 | β-Citronellal | terpene | floral/citrusy | 4 | 16 |
| 9 | 1589 | Unknown II | | cooked/vegetable | 64 | 64 |
| 10 | 1628 | γ-Butyrolactone | lactone | caramel/creamy | 64 | 128 |
| 11 | 1664 | 5-Methyl-2(5H) furanone | furan | burnt/roasted | 128 | 16 |
| 12 | 1867 | 2-Phenyl ethanol | alcohol | floral/rose | 128 | 512 |
| 13 | 1877 | Benzyl alcohol | alcohol | leaf/floral | 16 | 32 |
| 14 | 1905 | Piperitenone | ketone | sweet/floral | 128 | 512 |
| 15 | 1958 | Phenol | phenol | phenolic | 32 | 4 |
| 16 | 2037 | 2-pyrolidinone | pyrrole | phenolic/chemical | 64 | 32 |
| 17 | 2158 | Nonanoic acid | acid | green | 32 | 64 |
| 18 | 2287 | 3-Hydroxy-2-butanone | ketone | buttery/fatty | 8 | 16 |
| 19 | 2325 | 5-Hydroxy-2(3H) benzofuranone | furan | burnt | 64 | 16 |

[a] Linear retention index (LRI) calculated on DB-WAX capillary column. [b] FD factor of odorant compound was evaluated by AEDA.

*3.4. Odour Sensory Profiles of Teas Samples*

The means of numerical data of 11 attributes (color, aroma, fruity, green, astringency, herbal, floral, sweet, taste, bitterness, and general acceptability) obtained from the panelists during the sensory analysis of Moringa and Kinkeliba teas infused at cold and hot temperatures are displayed on the spider graph shown in Figure 3 for improved visualization. The teas from both samples were separately shown to the panelists to characterize the taste and

odor properties of tea infusion. Significant differences ($p < 0.05$) were found between cold- and hot-brewed Moringa and Kinkeliba tea. The sensory evaluation results were related to GC-O findings (Tables 3 and 4). The panelists preferred Moringa and Kinkeliba teas in hot tea form as they had higher general acceptability, aroma, and taste scores.

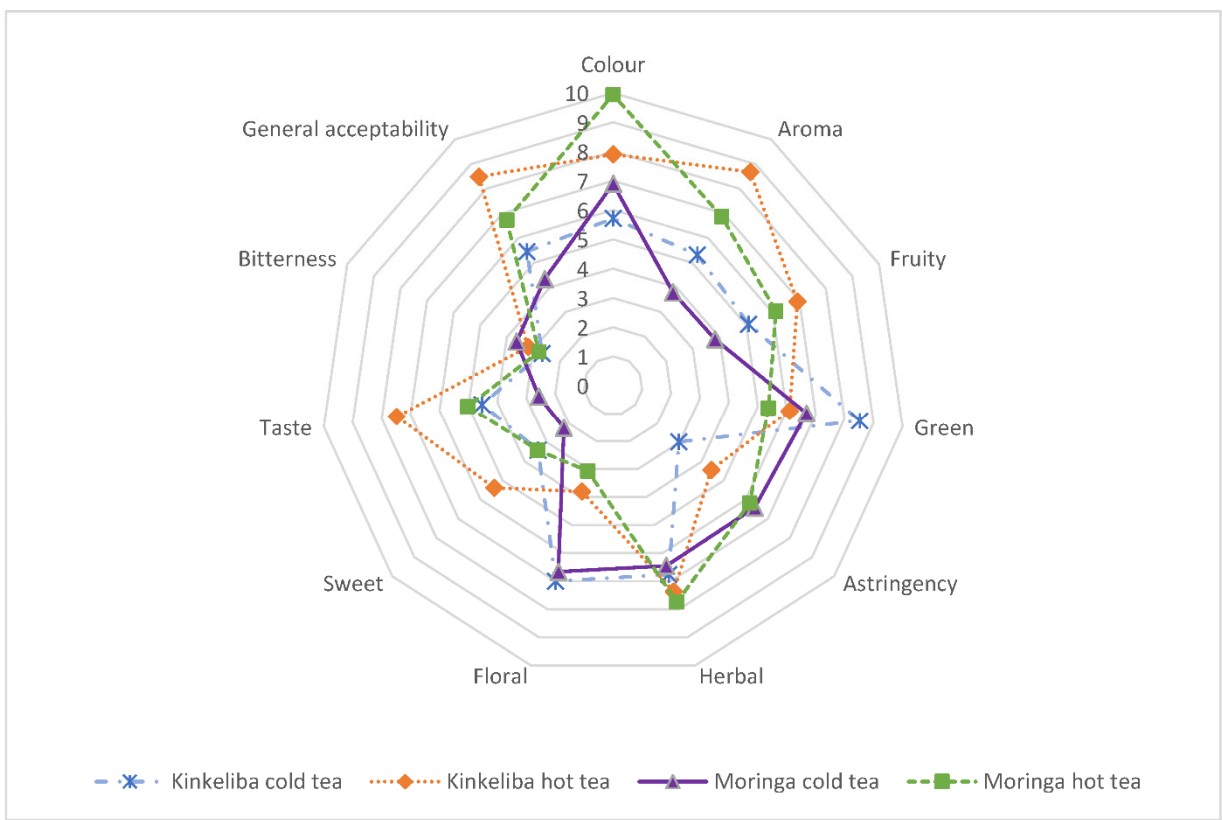

**Figure 3.** Sensory profiles of Moringa and Kinkeliba tea infusions.

As shown in Figure 3, both hot and cold infusions of Moringa tea appeared to be perceived as somewhat astringent, aromatic, and green with almost no bitterness. Hot-brewed Moringa tea displayed significantly higher scores in comparison to cold-brewed tea. Hot Moringa tea had an herbal note (7.74) with the highest color intensity (9.94), aroma note (6.87), astringency note (6.18), and fruity note (6.12). In the case of Moringa tea infused under cold brewing, it had higher astringency (6,41), floral (6,66), and green (6,68) notes. This was in agreement with GC-O results as FD values of compounds responsible for green, herbal odors, such as 3-hexanol, *(E)*-2-hexanal, and *(E)*-3-hexen-1-ol, were higher than hot brewed Moringa tea.

In the case of Kinkeliba, panelists defined the sensory attributes of hot and cold brewed teas as aromatic, fruity, slightly green, and sweeter. Kinkeliba tea samples infused at hot temperature had the highest general acceptability score (8.49) of all four samples in addition to higher aroma (8.68), fruity (6.93), herbal (7,37), and taste (7.48) scores. The green note (8.52) was perceived more in Kinkeliba tea brewed in cold conditions compared to the one infused in hot.

Consistent with GC-O analysis results, the green odor note had strong intensity in the sensory analysis of cold-brewed Kinkeliba tea (Figure 3). As shown in Table 4, the highest FD factors were recorded for 2-hexanol, 3-penten-2-ol, and 1-penten-3-ol compounds that were responsible for the green, grassy, and herbal notes and were highlighted in sensory evaluation of cold brewed Kinkeliba tea. The intensities of aroma, fruity, herbal, and sweet odors in hot brewed Kinkeliba tea were higher in the sensory analysis, similar to GC-O

analysis results. These findings confirmed that the aroma-active volatiles identified by GC-MS-O played a crucial role in creating the distinct aroma of Moringa and Kinkeliba teas.

## 4. Conclusions

Moringa and Kinkeliba herbal tea were obtained by two different infusions, hot ($98 \pm 2$ °C/10 min) and cold (room temperature; $25 \pm 2$ °C/24 h) techniques. The total phenolic content, antioxidant potential, aroma, and key odorant compounds of the obtained teas were assessed. This was the first in-depth research on the effects of different infusion conditions on these properties of Moringa and Kinkeliba herbal teas. The experimental data demonstrated that significant alterations occurred in the aroma and key odorant profile, total phenolics, and antioxidant activity of both tea samples due to the infusion conditions. The highest amount of total phenolic content and antioxidant capacity in both tea samples were found in hot brewed teas. On the contrary, aroma compounds were found in higher amounts in cold-brewed teas. For key odorants, Moringa and Kinkeliba teas had a similar pattern of aroma-active compounds, which was mainly dominated by alcohols and furans. Among the detected alcohols, *(E)*-3-hexen-1-ol (green/floral), 3-hexanol (green/floral), and 2-phenyl ethanol (floral/rose) were the most potent odorants with the highest FD factors in Moringa tea. FD values of these compounds were higher in cold-brewed teas. Kinkeliba tea samples' dominant aroma-active compounds were alcohols and furans. Among 19 key odorant compounds characterized in Kinkeliba tea samples, 3-penten-2-ol (herbal/green), 2-hexanol (green), and *(E)*-3-hexen-1-ol (green/floral) were revealed as powerful odorant compounds with the highest FD factors in the cold brewing method. The data obtained by GC-O analysis were also in agreement with the sensory evaluation results of the tea samples.

Therefore, when the infusion effect was evaluated, cold infusion ($25 \pm 2$ °C/24 h) was found to preserve the aroma of Moringa and Kinkeliba teas more, while hot infusion ($98 \pm 2$ °C/10 min) resulted in higher amounts of total phenolic content and antioxidant activity. Thus, our findings offer important information about the selection of infusion conditions when consuming Moringa and Kinkeliba teas.

**Author Contributions:** Conceptualization, S.S.; Software, O.S.; Validation, G.G. and M.S.; Formal analysis, K.F.O. and O.S.; Investigation, H.K.; Data curation, K.F.O., G.G., O.S., H.K. and S.S.; Writing—original draft, O.S.; Writing—review & editing, H.K., M.S. and S.S.; Visualization, H.K., M.S.; and Supervision, S.S. All authors have read and agreed to the published version of the manuscript.

**Funding:** This research received no external funding.

**Data Availability Statement:** Research data are not shared.

**Acknowledgments:** Authors thank Cukurova University Central Laboratory (CUMERLAB)-Turkey for the GC-MS analysis.

**Conflicts of Interest:** The authors declare no conflict of interest.

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
