# Peer review of "GC-MS-Olfactometric Characterization of Volatile and Key Odorants in Moringa (Moringa oleifera) and Kinkeliba (Combretum micranthum G. Don) Herbal Tea Infusions Prepared from Cold and Hot Brewing"

_separations, doi:10.3390/separations10010010_

Round 1

Reviewer 1 Report

The manuscript entitled «GC-MS-Olfactometric characterization of volatile and key odorants in Moringa (Moringa oleifera) and Kinkeliba (Combretum micranthum G. Don) herbal tea infusions prepared from cold and hot brewing» is the first attempt to conduct a comprehensive assessment of odor and taste of the investigated herbal teas and its correlation to the chemical composition. There are several strong and several weak points of this manuscript.

Strong points:

·       The information presented is new and useful for future study;

·       The data is adequately presented, figures and tables are presented in comprehensible format;

·       Chemical compositions of the infusions were determined by means of GC-MS with LRI and MS data, and in most cases by comparison with external standards;

·       GC-O data is in good agreement with the sensory evaluation;

·       Quantitative data is obtained via external standard (MS data?) and QAMS methods (using FID responses). However, it would be better to provide some more details;

·       Good comparison is made to the existing literature.

Weak points:

·       English should be significantly improved with assistance of the scientific English language expert; pay special attention to the abstract.

·       There are a lot of minor inadvertencies, such as mL, ml, +-0.00, etc;

·       GC-O and GC-MS-O, GC-MS-Olfactometry are used unsystematically; GC-FID is used once in a whole manuscript (in experimental part), what results were obtained by GC-FID?

·       PCA results seem hulking. There were only two 1.5 kg samples of two kinds of tea. What is the need to use multivariate statistics in this case? Simple chart diagrams could illustrate the differences in the chemical concentrations of volatile chemicals.

Additional questions:

According to the lines 423-425 it was possible to determine the odour of those compounds that were not identified due to the trace level. Was the same concentration (dilution) used for MS detection and Olfactometric characterization?

What reasons could stand for pH change in hot infusion of Moringa tea?

What was the purpose of citing ref. [28]? Does it describe the extraction procedure adopted by the authors? Extraction was a crucial step in this study, more details should be provided.

In general, I believe that this work can be reconsidered after revision.

Author Response

Addressing to Reviewer’s Comments

REVIEWER 1

  • English should be significantly improved with assistance of the scientific English language expert; pay special attention to the abstract.

Thank you for the contribution. The paper has been carefully revised by a native English speaker to improve the grammar and readability again.

  • There are a lot of minor inadvertencies, such as mL, ml, +-0.00, etc;

All units and symbols in the text were checked and corrected again. Please see related parts.

  • GC-O and GC-MS-O, GC-MS-Olfactometry are used unsystematically; GC-FID is used once in a whole manuscript (in the experimental part), what results were obtained by GC-FID?

Each of the detectors was used in order to carry out the identification, quantification and olfactometric analyses of aroma & aroma-active compounds. The identification of each volatile was performed using MS while quantification of each compound was done by integrating peak areas using FID signals and the Olfactometry was used to determine which compounds were aroma-active and possessed an impact on the overall aroma of tea infusion samples. 

  • PCA results seem hulking. There were only two 1.5 kg samples of two kinds of tea. What is the need to use multivariate statistics in this case? Simple chart diagrams could illustrate the differences in the chemical concentrations of volatile chemicals.

Thank you for your remark. You are absolutely correct, two samples are less in order to have an explainable PCA plot, However,  PCA permitted a better visualization of the variability of the data set and also provided clear proof of the structural interdependence of aroma compounds.

 Additional questions:

According to the lines 423-425 it was possible to determine the odour of those compounds that were not identified due to the trace level. Was the same concentration (dilution) used for MS detection and Olfactometric characterization?

Yes, the amount of the injected volume was same in all GC detectors. 3µl of the injection volume in the GC effluent was split 1:1:1 among the FID, MS, and sniffing port via a Dean’s switch. This split allowed us a simultaneous monitorization of FID signal for the quantification, an MS signal for the identification, and the odour characteristics of each compound detected by the sniffing port.

What reasons could stand for pH change in hot infusion of Moringa tea?

Actually, the change in pH level of hot infusion of Moringa tea was pointed out but not highlighted in the discussion part. The main difference could be sourced from the varietal differences and the different behaviors of specific phenolics under such challenging conditions.

What was the purpose of citing ref. [28]? Does it describe the extraction procedure adopted by the authors? Extraction was a crucial step in this study, more details should be provided.

Yes, it’s correct. The extraction of volatile compounds was explained in detail in the paper cited as reference [28] in this manuscript.

In general, I believe that this work can be reconsidered after revision.

Reviewer 2 Report

The English stile and the topic of the sentences is strange. It is recommended to make some English correction, if possible by a native speaker.

Line 13 – the email of corresponding author is written with another stile that the required

Line 19 – what is 10 mn? If is minutes, it should be 10 min

Line 20 – the word “while” is not necessary

Line 49 – the references mentioned [3-5] are about antioxidant and antimicrobial activity. What about the others mentioned in lines 45-48? “Recently, the clinical investigations proved that Moringa had enormous and effective health properties including anti-inflammatory, hyperglycemic alternative, weight regulation, anti-tumor, cardiac and circulatory stimulant, anti-diabetic, antipyretic, anti-epileptic, anti-ulcer, anti-bacterial, anti-fungal, anti-hypertensive, cholesterol-lowering”? . Please reference all the statements with references that can support them or delete those parts that are not proper referenced.

Same comment for the references [9-11] mentioned in line 59. If I check those references they are about antidiabetic, antibacterial properties, but in the text is written about “s tranquillizer, tonic, digestive and diuretic”.

Same comment for the ref mentioned in line 62. Just as a curiosity, which one from the authors is French speaker, because they included as ref article in French language.

Please clarify the points mentioned above and check all references one by one, those mentioned by me, and the other not mentioned.

Line 73 please replace “disappeared” with lost

Lines 98-99 – were the species identified by a specialist? Is there any specimen number deposited somewhere?

Line 127 - please change “obscurity” with dark

Line 153 – why the authors decided to start the temperature program from 60 degrees for such volatile constituents?

Line 155 – “A quantity of 3 μl extract was introduced into the device”. Please be more specific and scientific. Instead of introduced can be written injected and instead of device can be written GC injection port for example.

Line 159 “The injector type and heat settings had the same parameters as in gas chromatography”. Not sure what you want to say here

Lines 166-168 – please explain how exactly the quantification was calculated

Line 168 – the authors wrote “Each analysis was performed with 3 replications [21,23].” Why they put there 2 references?

Table 2 is very big. Could you please condense it? It seems to have a lot of useless space between the lines. If not, consider moving it in supplementary material.

Lines 324 – 340 – Reading all that text and observing the Figures 1 and 2 I can not say that the 2 infusion methods can be distinguished. There is not any obvious clustering and most of volatiles are populating the middle of graph. Could the authors mention what exactly they put in the PCA program to generate the graphs?

By the way the Figures are not nice designed. The text is either too evident, or too small. Please remove the bold and decrease the font of the factors (F1, F2…). Please do not include the Figure captions in the figure body.

I like the concept of Figure 3, but please remove the caption from the Figure

Author Response

Addressing to Reviewer’s Comments

REVIEWER 2

The English stile and the topic of the sentences is strange. It is recommended to make some English correction, if possible by a native speaker.

Thank you for the contribution. The paper has been carefully revised by a native English speaker to improve the grammar and readability again.

Line 13 – the email of corresponding author is written with another stile that the required

This part is corrected, please see the line 13.

Line 19 – what is 10 mn? If is minutes, it should be 10 min

The misspelling is corrected.

Line 20 – the word “while” is not necessary

The sentence is revised.

Line 49 – the references mentioned [3-5] are about antioxidant and antimicrobial activity. What about the others mentioned in lines 45-48? “Recently, the clinical investigations proved that Moringa had enormous and effective health properties including anti-inflammatory, hyperglycemic alternative, weight regulation, anti-tumor, cardiac and circulatory stimulant, anti-diabetic, antipyretic, anti-epileptic, anti-ulcer, anti-bacterial, anti-fungal, anti-hypertensive, cholesterol-lowering”? . Please reference all the statements with references that can support them or delete those parts that are not proper referenced.

Thank you for the contribution. Related reference is corrected. Please see it in References part.

Same comment for the references [9-11] mentioned in line 59. If I check those references they are about antidiabetic, antibacterial properties, but in the text is written about “s tranquillizer, tonic, digestive and diuretic”.

Thank you for the contribution. Related reference is substituted. Please see it in References part.

Same comment for the ref mentioned in line 62. Just as a curiosity, which one from the authors is French speaker, because they included as ref article in French language.

Related reference is substituted with another paper which is published in English. Thanks for the comment, one of our author is a native French speaker.

Please clarify the points mentioned above and check all references one by one, those mentioned by me, and the other not mentioned.

All references are re-checked and related parts are corrected.

Line 73 please replace “disappeared” with lost

The sentence is corrected.

Lines 98-99 – were the species identified by a specialist? Is there any specimen number deposited somewhere?

Related part is updated and some detailed information is provided. Please see it in “2.1. Plant materials and reagents”

Line 127 - please change “obscurity” with dark

The sentence is corrected.

Line 153 – why the authors decided to start the temperature program from 60 degrees for such volatile constituents?

Thanks for the disclosure. We corrected the mistake, initial temperature was 40°C indeed.

Line 155 – “A quantity of 3 μl extract was introduced into the device”. Please be more specific and scientific. Instead of introduced can be written injected and instead of device can be written GC injection port for example.

Related sentence and other similar points are revised.

Line 159 “The injector type and heat settings had the same parameters as in gas chromatography”. Not sure what you want to say here

Mentioned sentence is removed in order not the create confusion. In our configuration, MS, FID and Olfactometry were used at the same time on the same GC system using a Dean Switch apparatus which splits the flow into three equal volume as 1:1:1.

Lines 166-168 – please explain how exactly the quantification was calculated

Thank you for your valuable question. According to the Authors, to perform a semiquantitative determination, the use of a single internal standard that provided that a very good recovery yield of internal standard may be an appropriate approach. The recovery yield of the internal standard in one of our previous studies (Sonmezdag et al., 2018) was measured on six repeated extractions of pistachio oil and a yield of 95% was determined. The linearity and repeatability of the method were determined using calibration curves. Calibration curves were prepared for five main compounds through dilution of the volatile compound stock solution to five different concentrations with 5% (v/v) ethanol. Linearity was checked on standard solutions of those main constituents of the pistachio oil volatile fraction: in all cases, the r2 obtained by the linear regression equation was higher than 0.99. Repeatability of the semiquantitative determination of the volatiles having an average relative abundance higher than 1%, measured by six replicate extractions, was lower than 5%.

This explained semiquantitative determination of the absolute concentration of each volatile was also applied for the tea samples in current study. The amounts of each volatile were obtained by calculating its concentration in the extract as equivalent concentration of the internal standard, 4-nonanol, by using measured peak area ratio and known concentration of the internal standard.

Sonmezdag, A. S., Kelebek, H., & Selli, S. (2018). Pistachio oil (Pistacia vera L. cv. Uzun): characterization of key odorants in a representative aromatic extract by GC-MS-olfactometry and phenolic profile by LC-ESI-MS/MS. Food chemistry, 240, 24-31.

Line 168 – the authors wrote “Each analysis was performed with 3 replications [21,23].” Why they put there 2 references?

References were not provided at the right place. Related sentence is corrected by citing those references at the end of the previous sentence. Please see the related part.

Table 2 is very big. Could you please condense it? It seems to have a lot of useless space between the lines. If not, consider moving it in supplementary material.

The reviewer is right, thanks for the suggestion. Unnecessary spaces are removed from the table.

Lines 324 – 340 – Reading all that text and observing the Figures 1 and 2 I can not say that the 2 infusion methods can be distinguished. There is not any obvious clustering and most of volatiles are populating the middle of graph. Could the authors mention what exactly they put in the PCA program to generate the graphs?

Thank you for the valuable comment. The reviewer is correct, however, the effect of high infusion temperature on aroma profile of tea samples, which is the principal idea of this study, is quite clear considering the marker aroma compounds such as furfural, 2-acetylpyrrole, 2-pyrrolidinone 5-hydroxy-2(3H) benzofuranone and furfuryl alcohol. Indeed, the distribution of these compounds is clearly visible.

By the way the Figures are not nicely designed. The text is either too evident or too small. Please remove the bold and decrease the font of the factors (F1, F2…). Please do not include the Figure captions in the figure body.

Thanks for the suggestions. Figures are revised and bold characters are changed. Figure captions are provided in the required format of the “Separations” journal as explained in the journal guidelines.

I like the concept of Figure 3, but please remove the caption from the Figure

Figure captions are provided in the required format of the “Separations” journal as explained in the journal guidelines.

Round 2

Reviewer 1 Report

The authors have revised the manuscript in a proper way to address all recommendations made by the reviewers. Therefore, I can recommend it for publication in present form.

Author Response

Thank you for your acceptance.

Reviewer 2 Report

The authors answered most of my requirements and improved the manuscript. 

I will kindly ask them only to provide the voucher specimen number in the manuscript (page 3). This is the only one remaining comment that I have. 

Author Response

Addressing to Reviewer’s Comments

REVIEWER 2

I will kindly ask them only to provide the voucher specimen number in the manuscript (page 3). This is the only one remaining comment that I have. 

Response: Thank you for your suggestion. The voucher specimen number for each herb was added to page 3.